# Detection of Aspartylglucosaminuria Patients from Magnetic Resonance Images by a Machine-Learning-Based Approach

**DOI:** 10.3390/brainsci12111522

**Published:** 2022-11-10

**Authors:** Arttu Ruohola, Eero Salli, Timo Roine, Anna Tokola, Minna Laine, Ritva Tikkanen, Sauli Savolainen, Taina Autti

**Affiliations:** 1HUS Medical Imaging Center, Radiology, University of Helsinki and Helsinki University Hospital, P.O. Box 340, FI-00290 Helsinki, Finland; 2Department of Neuroscience and Biomedical Engineering, Aalto University, P.O Box 11000, FI-02150 Espoo, Finland; 3Department of Child Neurology, Helsinki University Hospital and Helsinki University, P.O Box 900, FI-01400 Vantaa, Finland; 4Institute of Biochemistry, Medical Faculty, University of Giessen, Friedrichstrasse 24, 35390 Giessen, Germany; 5Department of Physics, University of Helsinki, P.O. Box 64, FI-00014 Helsinki, Finland

**Keywords:** magnetic resonance imaging, aspartylglucosaminuria, lysosomal storage disorders, classification, supervised learning, thalamus

## Abstract

Magnetic resonance (MR) imaging data can be used to develop computer-assisted diagnostic tools for neurodegenerative diseases such as aspartylglucosaminuria (AGU) and other lysosomal storage disorders. MR images contain features that are suitable for the classification and differentiation of affected individuals from healthy persons. Here, comparisons were made between MRI features extracted from different types of magnetic resonance images. Random forest classifiers were trained to classify AGU patients (*n* = 22) and healthy controls (*n* = 24) using volumetric features extracted from T1-weighted MR images, the zone variance of gray level size zone matrix (GLSZM) calculated from magnitude susceptibility-weighted MR images, and the caudate–thalamus intensity ratio computed from T2-weighted MR images. The leave-one-out cross-validation and area under the receiver operating characteristic curve were used to compare different models. The left–right-averaged, normalized volumes of the 25 nuclei of the thalamus and the zone variance of the thalamus demonstrated equal and excellent performance as classifier features for binary organization between AGU patients and healthy controls. Our findings show that texture-based features of susceptibility-weighted images and thalamic volumes can differentiate AGU patients from healthy controls with a very low error rate.

## 1. Introduction

Aspartylglucosaminuria (AGU, OMIM 208400) is a rare lysosomal storage disorder caused by the deficiency of aspartylglucosaminidase (AGA), a hydrolase involved in lysosomal degradation of N-glycosylated proteins (reviewed in [1,2]). A progressive learning disability is observed in AGU patients from childhood onward. In addition, AGU patients manifest with recurrent infections, connective tissue and skeletal abnormalities, and behavioral problems. A fraction of AGU patients also has epilepsy. Although born seemingly normal, young adult AGU patients are severely handicapped with intellectual disabilities and require help to master daily life. Most AGU patients are located in Finland, but recently, increasing numbers of non-Finnish patients have been diagnosed. Because AGU is very rare outside Finland, the diagnosis of non-Finnish patients is often delayed.

Although no approved therapies for AGU are available, preclinical research in gene therapy and pharmacological chaperones has been carried out in recent years [3,4]. Therefore, it is increasingly important to identify patients who would benefit from therapies that are likely to be available within a few years. For this, novel diagnostic means based on radiological imaging approaches would be desirable.

AGU is normally diagnosed by either a direct genetic analysis of the *AGA* gene or the presence of aspartylglucosamine in a urine oligosaccharide screening, which is followed by genetic confirmation. The latter is the most common way of diagnosing AGU in Finland, and diagnosis based on MRI findings is rare. However, a brain MRI may have already been performed for many patients due to developmental delay or other indications before confirmation of an AGU diagnosis by genetic testing. As thalamic changes typical for AGU are also observed in young AGU patients [5,6,7,8], pre-existing images could be used to obtain the first diagnostic signs for an AGU diagnosis, especially with the help of a suitable means for automated image analysis. The availability of MR imaging has generally increased in many countries during recent years, and the short imaging sequences that have been developed decrease the need for anesthesia in special groups and small children. Therefore, MRI may provide an attractive alternative for the diagnosis of AGU.

The accumulation of paramagnetic compounds in the thalamic nuclei of AGU patients was previously identified based on a decreased signal intensity in filtered-phase susceptibility-weighted (SWI) MR images [5]. Furthermore, building on this finding, Sairanen et al. [6] showed that the paramagnetic compound accumulation in AGU-affected thalami strongly correlated with the patient’s age and the degree of disease progression. Neither study detected paramagnetic compound accumulation in the control group [5,6]. It should be noted that, depending on the handedness of a magnetic resonance imaging scanner, paramagnetic compounds appear either brighter (left-handed scanner) or darker (right-handed scanner) in a filtered-phase SWI [5]. In magnitude SWI (left-handed scanner), the accumulation appears as a decreased signal intensity [6]. The signal intensity decrease has also been detected in the thalami of AGU patients in T2-weighted images. The decrease was reported to be more pronounced in the pulvinar nuclei of the thalamus [7,8]. Similar MRI findings have also been reported in other lysosomal storage diseases. In more than 40 classified lysosomal storage diseases, some degree of decreased T2 signal has been reported [9]. The signal changes are thought to be caused by excessive metabolite accumulation that increases the viscosity, leading to T1 and T2 shortening [9].

AGU is also known to have a diminishing effect on the volume of the thalamus [10]. Tokola et al. [10] showed that the thalamic volume of a 9.9-year-old boy with AGU was 28.8% smaller than the thalamic volume of his dizygotic, healthy twin brother. To pave the way for computer-assisted diagnostic tools that automatically analyze MR images and differentiate individuals with AGU, and possibly other lysosomal storage disorders, from the healthy population, it is essential to first study which imaging sequences and features are the most robust for a simple binary classification. The development of such tools is important because human experts may miss the diagnosis of rare diseases in populations with low incidences. The previous literature on AGU was screened for MRI findings and suitable features for which statistically significant differences were observed between AGU patients and healthy controls in quantitative analyses. Three findings, i.e., signal changes in SWI and T2-weighted MR images and reduced thalamic volume, constitute the features used here for a machine-learning-based classification.

Previous findings on applying machine-learning principles to medical images of individuals affected with lysosomal storage diseases are scarce. Sharma et al. [11] used a machine-learning model based on stereological and texture features in micro-MRI images to discern between patients suffering from another lysosomal storage disease, type 1 Gaucher disease, and healthy controls, with a maximum accuracy of 73% and 0.79 area under the curve (AUC). The application of data mining and machine learning to search for new inhibitors and chaperones to generally treat lysosomal storage diseases was suggested by Klein et al. [12]. We provide a machine-learning-based method capable of distinguishing AGU patients from healthy controls based on analysis of specific features in MR images. Previously, similar features were used in supervised learning approaches to differentiate cognitive disorders [13] and identify early cognitive impairment [14].

## 2. Materials and Methods

### 2.1. Subjects

In total, the data of 46 unique subjects were used for the analyses in this study. The dataset consisted of 22 patients with an AGU diagnosis and 24 healthy volunteers as controls. Table 1 shows further details (sex and age) of the subjects, sorted by image type. Written informed consent was obtained from the parents of the subjects.

### 2.2. MRI Sequence Parameters

The susceptibility-weighted images used in this study were acquired using a 3T MAGNETOM Skyra (Siemens, Erlangen, Germany) scanner. The imaging parameters of the SWI sequence were a TR of 27 ms, TE of 20 ms, flip angle of 15°, slice thickness of 2.0 mm, and in-plane resolution of 0.86 × 0.86 mm, with a matrix size of 256 × 232. Some of these images were previously used in Sairanen et al. [6] and Tokola et al. [5]. SWI acquisition was executed on a left-handed MR scanner [6].

The T1- and T2-weighted images were acquired using a 3T MAGNETOM Skyra (Siemens, Erlangen, Germany). T2 axial series (TR of 4000 ms, TE of 82 ms, slice thickness of 3 mm, in-plane resolution of 0.4978 mm × 0.4978 mm, flip angle of 150°, and matrix of 448 × 448) and T1 3D-MPRAGE series (TR 2000 ms, TE 2.74 ms, slice thickness 1 mm, in-plane resolution 1 × 1 mm, flip angle 10°, and matrix size 256 × 256) were acquired. This material was previously used by Tokola et al. [8].

### 2.3. Image Segmentation and Feature Extraction

For each subject, the T1-weighted anatomical images were preprocessed and volumetrically segmented using the FreeSurfer (https://surfer.nmr.mgh.harvard.edu/) (accessed on 7 September 2022) cortical reconstruction process recon-all [15,16,17,18,19,20,21]. Thereafter, the thalamus of each subject was further segmented into 25 separate nuclei (left and right) using the FreeSurfer tool segmentThalamicNuclei.sh [22]. The thalamic nuclei segmentation tool utilizes a probabilistic atlas, built using ex vivo MRI brain scans and histological data, to produce parcellation of the thalamus into 25 thalamic nuclei [22]. The estimated volumes of the nuclei, obtained as the tool’s output, were left–right-averaged to obtain the 25 volumes used as features for the classification. All volumetric measures used to train the classifiers were normalized by dividing them by the total estimated intracranial volume. The total intracranial volumes were estimated using the recon-all pipeline [15,16,17,18,19,20,21]. The T2-weighted images and SWI were registered with T1-weighted images using 3DSlicer (4.8.0), a rigid transformation model, and a mutual information criterion [23]. The registration was conducted to mitigate the possible effects of head translation and rotation between the imaging sequences. The whole thalamus volume estimated by the nucleic segmentation tool and the thalamus volume output provided by the subcortical segmentation pipeline of the Freesurfer (also referred to as aseg segmentation) were used to create two feature matrices to inspect and contrast the robustness of the nucleic-volume-based classification. In summary, three feature matrices were obtained: the first contained 25 left–right-averaged nucleic volumes, the second contained the whole volume estimated by the nucleic segmentation tool, and the third contained the whole volume estimated by the subcortical segmentation pipeline. The discrete segmentation volumes in the FreeSurfer voxel space for each subject output by the thalamus segmentation tool were binarized into masks that were then used to extract the thalamus from the SWI. The PyRadiomics package [24] was used to extract radiomics features from the SWI of the thalami. The gray level size zone matrix zone variance (GLSZM zone variance or zone variance) was chosen as the radiomics feature used for the classification. The gray level size zone matrix [25] is a statistical matrix used for texture classification. In a gray level size zone matrix, *P(i, j)*, the *(i, j)*th element of the matrix is the number of connected voxels of the gray level intensity, *i*, and the size, *j* [26]. Voxels of the same gray level intensity are considered connected in 3D if they are connected, i.e., the distance between the voxels according to the infinity norm is 1 [26]. Zone variance (*ZV*) is then calculated by:$$ZV=\overset{N_{g}}{\underset{i=1}{\sum }}\;\overset{N_{s}}{\underset{j=1}{\sum }}p\left(i,j\right){\left(j-\mu \right)}^{2}$$
where *P*(*i*, *j*) is the normalized gray level size zone matrix defined as $p\left(i,j\right)=\frac{P\left(i,j\right)}{N_{z}}$, *N_s_* is the number of discreet zone sizes in the image, *N_g_* is the number of discreet intensity values in the image, and $\mu =\overset{N_{g}}{\underset{i=1}{\sum }}\;\overset{N_{s}}{\underset{j=1}{\sum }}p\left(i,j\right)j\;$ [26]. Default gray level discretization was used.

The caudate nucleus segmentation output by the recon-all FreeSurfer pipeline was used to extract the caudate nuclei of each subject from the T2-weighted MR images. The average intensity value was calculated after eroding the mask, which was performed to avoid including high-intensity cerebrospinal fluid signals from the borders due to a partial volume effect and possible inaccuracies in the segmentation and registration processes. For each subject, the average intensity of the caudate nuclei was then divided by the average intensity of the thalamus, extracted from the T2-weighted images with an eroded binary mask. This calculation of caudate–thalamus mean intensity ratios yielded another feature matrix used for the classification.

Figure 1 shows example data and the segmentation masks used to extract the classification features.

### 2.4. Random Forest Classifier

To test and compare the robustness of the extracted features for the classification between healthy controls and AGU patients, the MATLAB (The MathWorks Inc.:, Natick, MA, USA) random forest classifier function TreeBagger was used to grow bootstrap-aggregated decision trees. The decision trees were grown using five separate feature matrices: one containing 25 left–right-averaged, normalized nucleic volumes of the thalami; a second containing total normalized thalamic volumes estimated by the thalamus segmentation tool; a third containing the whole normalized thalamic volumes estimated by the recon-all pipeline; a fourth containing the zone variances of the thalami from SWI; and a fifth containing the caudate–thalamus mean intensity ratios from the T2-weighted images. Table 2 shows the number of data points used per each model. Table 3 shows the TreeBagger function parameters that were used. The cost was chosen so that a false negative was five times more costly than a false positive because missing the diagnosis of a patient suffering from a serious illness is usually a graver mistake.

Due to the limited available data, the classifiers were evaluated using a leave-one-out cross-validation technique [27]. Decision trees were grown in a leave-one-out fashion, where one subject was left out from the training, and the resulting model was used to classify the left-out subject. This process was repeated so that each subject was left out of the training and then classified. The performance of the classifiers was evaluated using the area under the receiver operating characteristic curve, accuracy, sensitivity, and specificity as metrics.

## 3. Results

Caudate–thalamus intensity ratios in T2-weighted images were calculated as previously described and plotted against the nucleic volume sum and age. Figure 2 shows the relationship between the total thalamus volume (as calculated by segmentThalamicNuclei.sh) and the caudate–thalamus intensity ratio in T2 images. AGU patients were more distinctly separated by the nucleic volume sum of the *y*-axis than the intensity ratio of the *x*-axis. AGU patients (M = 1.29, SD = 0.06) and controls (M = 1.21, SD = 0.03) exhibited a statistically significant difference in the caudate–thalamus intensity ratio in the T2-weighted images with *p* < 0.001, using an alpha level of 0.05. The random forest classifier resulted in an AUC = 0.8008 when using the settings described in Table 3 and the caudate-thalamus intensity ratios calculated from the T2-weighted images as classification features.

To investigate the effects of age on the caudate-thalamus intensity ratio classifier feature, the intensity ratio was plotted against the subject’s age. Figure 3 shows a scatter plot of the caudate-thalamus intensity ratio in the T2-weighted images and the subject’s age in years. AGU patients and controls showed differing intensity ratio values. No statistically significant correlation was found between age and the caudate-thalamus intensity ratio for either subject group at the significance level of 0.05.

To compare and visually evaluate the performance of the three volume-based classifiers, receiving operating characteristics (ROC) curves were calculated and plotted. Figure 4 shows the ROC curves for the three volume-based random forest classifiers. The performance of the classifier Model 1 from Table 2 is shown in green, wherein 25 left-right averaged, normalized thalamic nucleic volumes were used as features. The area under the curve (AUC) was 0.9792, signifying that the classifier was robust in its performance. In Figure 4, the ROC curve of the classifier Model 2 from Table 2, where the sum of the nucleic volumes was used as the classification feature, is shown in blue. With an AUC of 0.9735, the performance of this classifier was also very robust. Models 1 and 2 had equal accuracies, sensitivities, and specificities, and their performance could be considered equal. The ROC of the volume-based classifier with the lowest performance, Model 3 from Table 2 (AUC 0.9622), is shown in red in Figure 4. It used normalized thalamic volume, as estimated by the subcortical segmentation pipeline of the Freesurfer (aseg), as the classification feature. Its performance was slightly lower than that of Models 1 and 2.

To visually inspect and compare the grouping of the data points, the zone variances and the sum of thalamic nucleic volumes were plotted as a scatter plot. Figure 5 displays the relationship between the thalamus nucleic volume sum of the subjects (*x*-axis) and the zone variance (*y*-axis). The data points of the control group are more spread out, whereas the data points of the AGU patients are clustered closer together. The two subject groups are almost completely spatially separated at both the *x*- and *y*-axes, suggesting the robustness of the zone variance parameter in the classification.

The effect of age on the zone variance classifier feature was further investigated. Figure 6 shows the scatter plot of the subject’s age in years (*x*-axis) and the zone variance parameter. The differences in zone variance in the two subject groups were not due to the age of the subjects. The zone variances of AGU patients and healthy controls did not correlate with age at the significance level of 0.05.

Table 4 shows the AUCs, accuracies, sensitivities, and specificities of the trained classifiers from Table 2. Measured according to the performance metrics of Table 4, the zone variance of SWI was the most robust feature for classification, with 100% sensitivity. The T2-weighted caudate-thalamus intensity ratio was the lowest-performing feature in every metric.

## 4. Discussion

The robustness of the volumetric features calculated from T1-weighted images, the zone variance calculated from susceptibility-weighted images, and the caudate–thalamus intensity ratio in T2-weighted images were evaluated for binary classification between patients with AGU diagnosis and healthy controls using a random forest classifier and leave-one-out cross-validation. The areas under the receiver operating characteristic curves, accuracies, sensitivities, and specificities were used to compare the performance of the classifiers based on different features. Regarding the three normalized thalamic-volume-based classifiers shown in Figure 3, the classifiers using 25 left–right-averaged, normalized thalamic nucleic volumes and normalized nucleic volume sums (Models 1 and 2 in Table 2, respectively) performed slightly better than Model 3 trained classifiers with normalized aseg volumes. Iglesias et al. [22] reported that the volumes of the thalamic nuclei estimated by the thalamus segmentation tool are more robust for binary classification of Alzheimer’s disease patients and healthy controls than the volumes from Freesurfer’s subcortical segmentation pipeline or the nucleic volume sum. We observed a slightly better classification performance using the normalized nucleic volumes compared with the normalized volumes from Freesurfer’s subcortical segmentation pipeline. With our small sample size, the slight improvement in classifier performance could be attributed to the randomness of the classifier.

With an AUC of 0.9922 for Model 4, the radiomics feature zone variance of the gray level size zone matrix calculated from the SWI, was a near-perfect classifier feature for binary classification between diagnosed patients and healthy controls. Even though iron accumulation in the thalamic nuclei was shown to strongly correlate and increase with age and disease progression in AGU [6], the zone variances of the gray level size zone matrices of the entire thalami of the subjects did not show a statistically significant correlation with age on a significance level. This finding implies that zone variance is a robust and clinically interesting classifier feature for subjects of all ages, at least when using the entire thalamic volume for the calculations. The clinical implication of the robustness of the SWI zone variance in classification is that the susceptibility-weighted imaging sequences should be favored in diagnostic practice. The T2-weighted caudate–thalamus intensity ratio based classifier (Model 5) was the weakest classifier, with an AUC of 0.8008. Considering this and the works of Sairanen et al. [6] and Tokola et al. [5], SWI sequences may be more useful for studying aspartylglucosaminuria than the T2-weighted sequences.

Potential limitations of our study are the low number of probands and patients used for training purposes. These limitations are due to the disease’s rareness and the differences in the MR images, which depend on the type of scanner used to obtain the images. Therefore, attention should be paid to the MRI scanner type and the technical specifications of the scanner when susceptibility-weighted images are analyzed, as compounds that distort the local magnetic field appear either as hypo- or hyperintensities, depending on the manufacturer. Future studies with MR images of AGU patients obtained with different scanner types but classified with our dataset will be useful to see if an improved training dataset will be required to classify images obtained using other scanners. However, regardless of the scanner manufacturer, the accumulated magnetic field distorting compounds are visible in SWI.

## 5. Conclusions

Our study demonstrates that even in a rare systemic disease such as AGU, machine-learning-based analyses can differentiate previously diagnosed AGU patients from healthy controls when models are trained with simple volume and texture features calculated from various types of magnetic resonance images. In this study, our previous MRI findings of thalamic abnormalities were confirmed. Furthermore, these thalamic signal intensity and volumetric abnormalities could distinguish AGU patients from controls. We could also dissect which abnormalities in the patient images were the most effective ones for classification. The thalamus thus plays an important role in lysosomal storage diseases. Computer-aided analyses are useful, especially in diagnosing rare systemic diseases seldom seen by radiologists. Larger data sets would be beneficial in creating improved machine-learning models with a lower risk of over-fitting.

## Figures and Tables

**Figure 1 brainsci-12-01522-f001:**
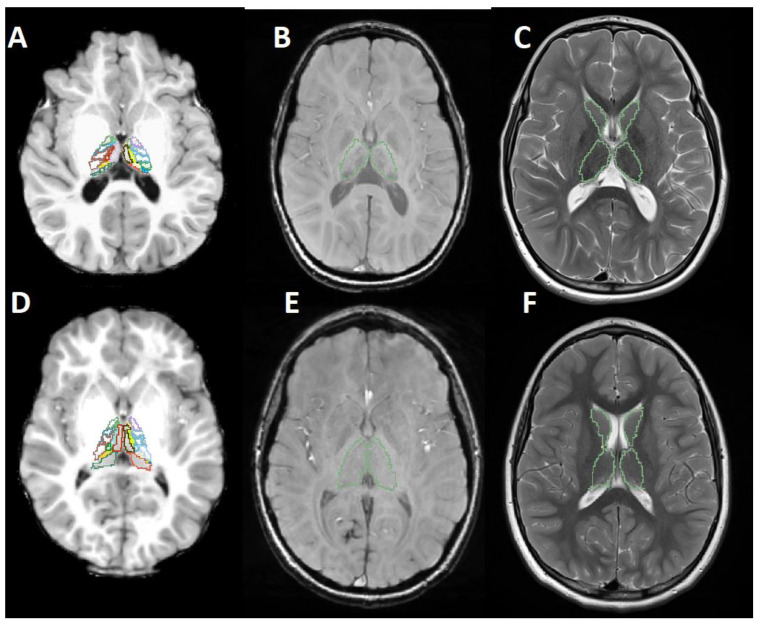
MR images of an AGU patient (**A**–**C**) and an age- and sex-matched healthy control (**D**–**F**). (**A**,**D**) T1-weighted image with segmented thalamic nuclei. (**B**,**E**) Susceptibility weighted image with the outlines of the segmented thalamus. (**C**,**F**) T2-weighted image with the outlines of the segmented thalamus and the caudate.

**Figure 2 brainsci-12-01522-f002:**
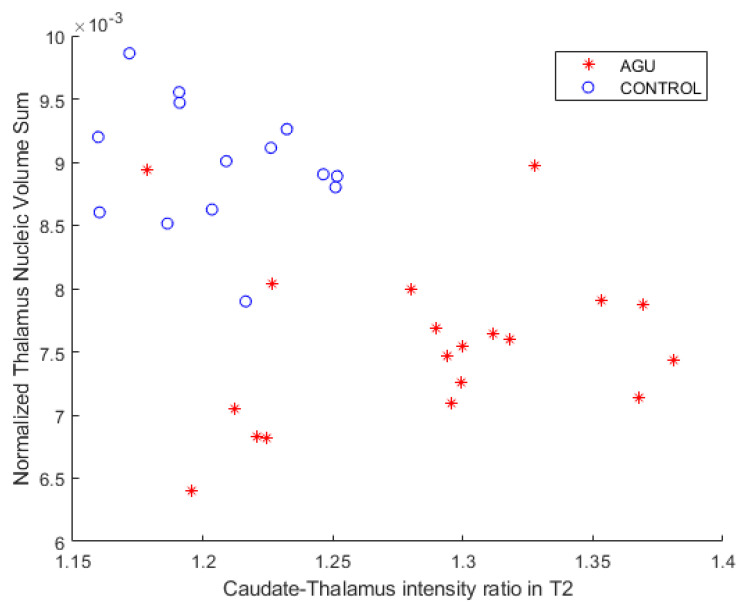
Normalized total thalamus volume (as sum of thalamus nucleic volumes) plotted against caudate-thalamus intensity ratio in the T2-weighted images. The patients and the controls were less separated by the *x*-axis than the *y*-axis.

**Figure 3 brainsci-12-01522-f003:**
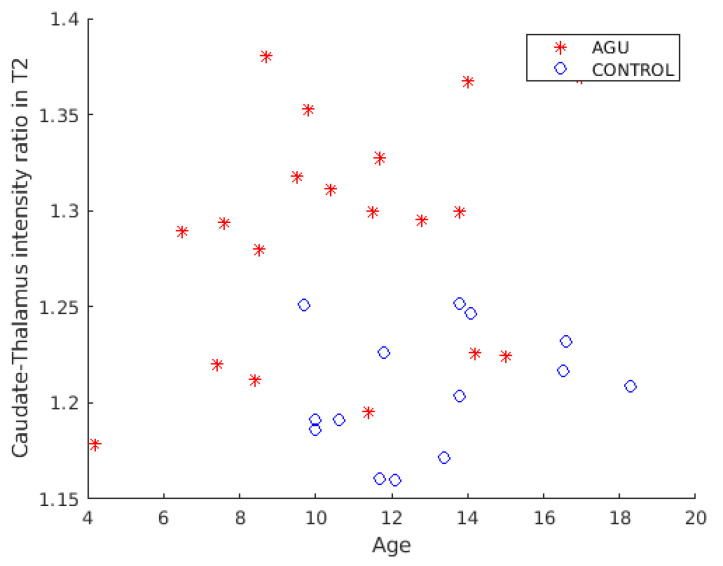
T2-weighted caudate-thalamus intensity ratio plotted against the age in years. The *y*-axis shows the caudate-thalamus intensity ratio in the T2-weighted images, whereas the *x*-axis depicts the age in years.

**Figure 4 brainsci-12-01522-f004:**
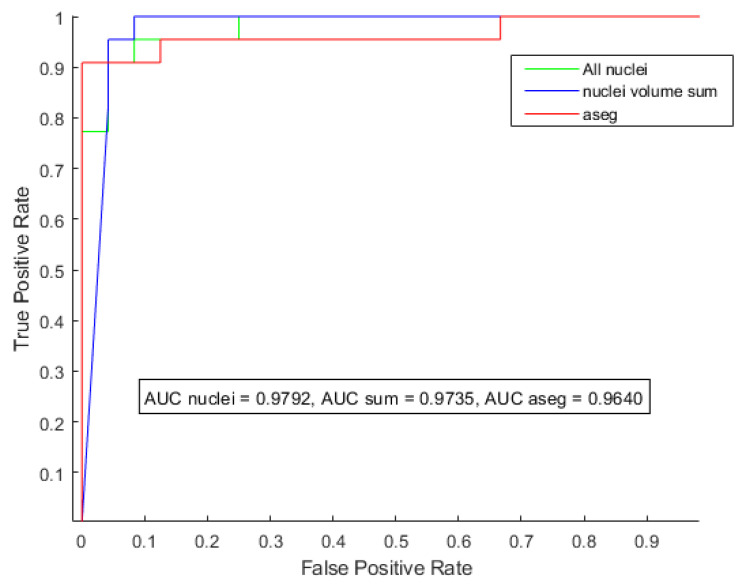
Receiver operating characteristic curves of the three classifiers for a positive diagnosis, predicted vs. actual. Green: the classifier using 25 left–right-averaged normalized thalamic nucleic volumes (Model 1). Blue: the classifier using normalized nucleic volume sum (Model 2). Red: the classifier using total normalized thalamic volume as estimated by the subcortical segmentation pipeline (Model 3).

**Figure 5 brainsci-12-01522-f005:**
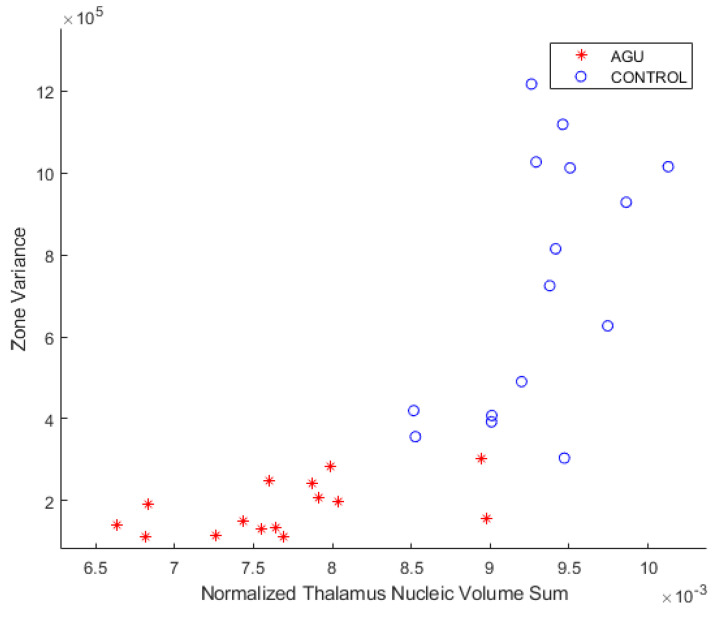
Zone variance calculated from the SWI, plotted against the sum of the thalamic nucleic volumes. Patient and control groups are separated in both *x*- and *y*-axes. Zone variance is on par with the thalamus volume as a classification feature.

**Figure 6 brainsci-12-01522-f006:**
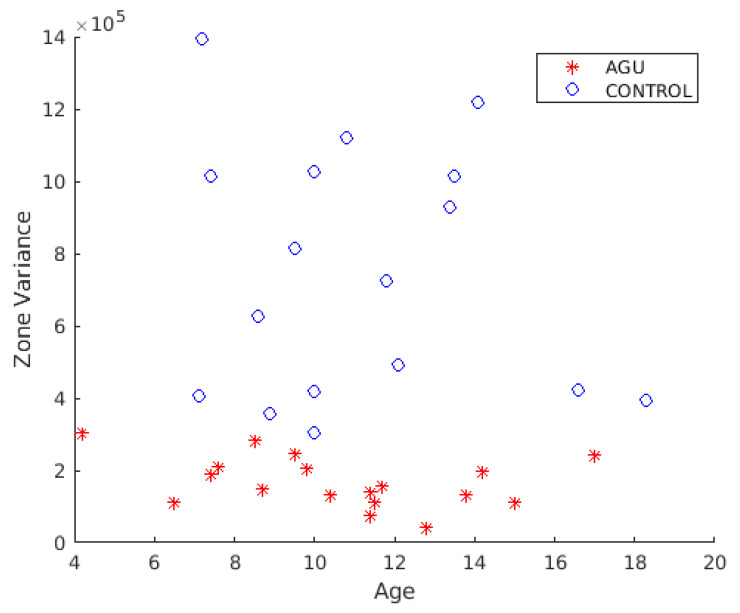
Subjects’ zone variance plotted against the age in years. No correlation with the age of the subject was found in either group at the significance level of 0.05.

**Table 1 brainsci-12-01522-t001:** Patient and control numbers, sex, and mean ages with standard deviation.

MR Image Type	Patient Number	Control Number	Mean Age Patient (years)	Mean Age Control (years)
T1	12 male	16 male	11.0 ± 3.4	10.7 ± 2.4
	10 female	8 female	10.1 ± 2.9	11.0 ± 2.1
	22 total	24 total	10.6 ± 3.1	11.2 ± 3.0
T2	12	5	11.0 ± 3.4	11.8 ± 2.0
	7 19	9 14	10.0 ± 3.3 12.8 ± 3.3	13.7 ± 2.9 13.0 ± 2.7
SWI	10	7	11.0 ± 3.4	8.7 ± 1.5
	6	9	10.1 ± 3.4	12.1 ± 2.8
	16	16	10.7 ± 3.3	10.8 ± 3.0

**Table 2 brainsci-12-01522-t002:** Details of the models and number of data points used for the training.

Model	MR Image Type	Features	Patients	Controls	Total Subjects
1	T1	Nucleic Volumes	22	24	46
2	T1	Whole Volume	22	24	46
3	T1	aseg Volume	22	24	46
4	SWI	Zone Variance	16	16	32
5	T2	Caudate-Thalamus Ratio	19	14	33

**Table 3 brainsci-12-01522-t003:** The MATLAB TreeBagger classifier parameters used.

Title 1	Title 3
NumTreees	4000
OOBPrediction	on
Method	classification
Options	statset(‘UseParallel’,true)
cost	[0 1; 5 0]

**Table 4 brainsci-12-01522-t004:** The performance metrics of the trained random forest classifiers.

Model	MR Image Type	Features	AUC	Accuracy	Sensitivity	Specificity
1	T1	Nucleic Volumes	0.9792	0.9348	0.9545	0.9167
2	T1	Whole Volume	0.9735	0.9348	0.9545	0.9167
3	T1	aseg Volume	0.9622	0.9333	0.9130	0.8750
4	SWI	Zone Variance	0.9922	0.9688	1.0000	0.9375
5	T2	Caudate–Thalamus Ratio	0.8008	0.6667	0.8947	0.3571

## Data Availability

The data are not publicly available due to privacy considerations of the patient data.

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
