# Peer review of "Detection of Aspartylglucosaminuria Patients from Magnetic Resonance Images by a Machine-Learning-Based Approach"

_brainsci, 2022, doi:10.3390/brainsci12111522_

Round 1

Reviewer 1 Report

The authors presented a machine learning classification scheme using several MRI modalities for the diagnosis of AGU. My comments are given below. 

1. Introduction could include recent ML approaches in this research area.  

2. Apart from MRI, what is the gold standard for diagnosing AGU? That should be included in the manuscript. What is the scale used for grading AGU? How did you grade AGU, binary or linear scale?

3. In-plane resolutions for T1 and T2 weighted images should be provided. 

4. What total features extracted from all three MRI modalities should be included in the form of a table.

5. It's good that you have used Freesurfer, however, a bit more details about pre-processing steps could be included such as reorientation, cropping, and bias field correction for the sake of a wide variety of readers. In my experience, FSL seems to be better for rigid/affine registration. Why did you conduct rigid registration of T2 and SWI to the corresponding T1 weighted image? It should be included for better clarity.

6. No sample images of the dataset are shown. 

7. There are many publicly available datasets that can be considered for the Healthy subjects pool such as ABIDE I and ABIDE II. You must consider increasing the data size. 

8. Why only one feature is selected from the GLSZM matrix?

9. Since you have only fewer subjects the actual values of the features can be given in the form of a table. 

10. Zone-weighted feature itself achieved 99.71% AUC. What is the use of other features?

11. Along with AUC, other performance metrics like accuracy, sensitivity, specificity, and precision scores should be given. 

12. Did you correct the volume values for head size? If not the statistics should be redone by normalizing the volumes with subject head size or total ICV.

13. Comparison with previous studies should be incorporated into why your method is superior. 

14. Considering that the subjects are kids around 10 years old, how do you make sure minimal subject movement during MRI acquisition? The acquisition time for each MRI protocol should also be given. 

15. Features from T1/T2 (myelin) mapping may have better contrast since you have both modalities available to you. 

16. There are many ML models other than random forest (RF) such as kNN, Naive Bayes, linear discriminant analysis, adaptive boosting, and support vector machines. Why are none of these explored? A thorough comparison of the proposed RF classifier with some of the aforementioned classifiers should be made. Otherwise, a rationale for choosing the RF classifier should be given. 

Overall, the manuscript should be significantly improved by increasing the data size and including several other performance metrics. Cross-validating and testing an ML classifier with such low data leads to biased results. The classier may miserly fail on unseen data.

Author Response

Dear Reviewer,

Thank you for your time dedicated to reviewing your manuscript, we have now revised the manuscript and present a response to your comments

Reviewer 2 Report

Reviewer comments for article brainsci-1950121 Identification of Aspartylglucosaminuria Patients from Magnetic Resonance Images by a Machine Learning-based Approach.

The authors present the differentiation of patients with aspartylglucosaminiuria against normal patients using MRI imaging and machine learning classification. The description of the methods, study design, and the implementation of random forest are well described.

However, I noticed three deficiencies and recommend several items that would help improve the manuscript.

Deficiencies

1.       It is unclear to the reader if it been previously established that the patterns described in this report can uniquely identify patients with aspartylglucosaminuria? If not, the use of “identification” should be amended throughout the manuscript because other disorders were not assessed. The use of “differentiate” in the first sentence of the conclusion appears to be more acceptable. The uniqueness of the features should also be clarified in describing previous findings (Lines 34-50).

2.       The introduction should acknowledge the use of clinical metabolic, enzymatic, and genetic testing for the detection and identification of patients with aspartylglucosaminuria. And, it may be possible that radiological imaging is typically more expensive and there is less appointment availability.

3.       The goal of identifying patients with aspartylglucosaminuria or other inborn errors of metabolism is to identify them when they are as young as possible. I acknowledge that finding neonates for this study would have been extremely difficult, but the study participants are in their teens. Can these patterns be detected in pre-symptomatic patients? This should be explained and transparently described, and amendments should be made in the introduction to better reflect the utility of this report.

Recommendations

1.       Exemplar MRI images would help visualize what exactly is being used for classification.

2.       I noticed that there were less participants in Models 4 and 5 (Table 2). Would that have any impact on the AUCs (Table 4)? Were any participants purposely excluded? Also, the abstract only contains the larger study sizes.

3.       The last two paragraphs in the introduction (lines 61-65) appear to be broken up by accident (this may be a formatting issue of the review PDF).

4.       The manuscript would be greatly improved by an improved results section. For example, lines 144- 156 read as bullet points.

Author Response

(The authors gave the same response as above.)

Round 2

Reviewer 1 Report

A few further comments below based on the author's response are below. 

The response to comment 10 is not clear. 

Table 4 caption is not clear. How do you define classifiers based on a single feature? 

For the SWI feature, both sensitivity and specificity are 1, whereas accuracy is 0.9688, consider correcting it. 

I can't see the performance of your proposed classifier. It should be included in Table 4. 

Images can be shown for both healthy and diseased. 

Any differences in the hippocampus, amygdala, and mamillary bodies between healthy and diseased?

Author Response

Dear Reviewer,

Thank you for your time dedicated to reviewing our manuscript, we have now made corrections to the manuscript

Reviewer 2 Report

In an effort to further improve this manuscript, I note the following deficiencies and recommendations.

Deficiencies

1. The title of the manuscript still contains "identification". Unless I have still misunderstood the introduction (lines 76 and 77) and data reported here, it would appear that a more appropriate word would be "detection". 

2. Lines 52 and 53: Is an MRI considered diagnostic? I believe the ACMG only considers results from gene sequencing as diagnostic. Positive metabolic results are not considered diagnostic.

3. Thank you for adding exemplar MRI images. The caption for Figure 1 needs to be improved. It is unclear if the MRI images are from a normal or an affected patient. To help the reader better appreciate your work, I also recommend placing both normal and affected images for comparison.

4. Please re-check the caption for Figure 4. The color labels do not appear to match. There does not appear to be a "black" line. Although this could be related to monitors, I checked this using several monitors.

Recommendations

A. Your reply "Elsewhere in the world, AGU is diagnosed even much later, in many cases only after the age of 10. Many of these patients may have already been subjected to MR imaging due to other indications, and it would thus be possible to use the imaging findings for an additional diagnostic measure" was essential in showing me why this MRI work is important. I was under the impression that children suspected with an inborn error of metabolism would get exome sequencing performed, perhaps simultaneously with an MRI. The importance of your explanation is not clearly communicated to the reader. I highly recommend streamlining the introduction to better highlight the unmet need this report addresses.

B. The manuscript may be stronger with MRI images as a Result. For example " the radiomics feature Zone Variance of the Gray Level Size Zone Matrix calculated from the SWI was found to be a nearly perfect classifier feature" (Lines 310 to 311). After looking at the MRI images, I do not know which part is the Zone Variance of the Gray Level Size Zone Matrix. Perhaps arrows and labels on the MRI would be beneficial for the reader.

C. In general, machine learning models "train" on a specific set of data generated using specific instrument parameters. Data classification is accurate only when using the same specific instrument parameters.  Are there any differences in MRI instruments/images among different MRI vendors that would prevent the use of your "training" data? Or would another institution with different MRI instruments have to create their own "training" images?

Author Response

(The authors gave the same response as above.)
